# Verhulst-type equation and the universal pattern for global population growth

**Agata Angelika Sojecka[1]\*, Aleksandra Drozd-Rzoska[2]\***

**1** Department of Marketing, University of Economics, Katowice, Poland, **2** Institute of High Pressure Physics Polish Academy of Sciences, Warsaw, Poland.

\* arzoska@unipress.waw.pl (AAS); agata.angelika.sojecka@gmail.com (AD-R)

## Abstract

The global population $P(t)$ (growth from 10,000 BCE to 2023) is discussed in frames of the Verhulst-type scaling, recalling the sustainable development concept. The analysis focuses on the per capita global population growth rate, for which the analytic counterpart is considered: $G_P(P) = \frac{\left(\frac{dP(t)}{P(t)}\right)}{dt} = \frac{dlnP(t)}{dt}$. The focused insight reveals two near-linear domains for $G_P(P)$ changes: from ~700 CE till ~1968 and from ~1968 till 2023. It can be considered a reference pattern for long-term global population changes. For models recalling the Verhulst-type scaling, such analysis indicates that a single pair of growth rate and system resource coefficients $(r, s)$ should describe the rise in the global population. However, the Verhulst relation with such effective parameters does not describe $P(t)$ changes, which raises the question of whether it is adequate to describe global population changes. Notably is the new way of data preparation, based on their collections from various sources and numerical filtering to obtain a 'smooth' optimal set. The changes of $P(t)$ were analyzed via the 'reversed protocol' analysis, in comparison to the standard pattern, namely: (*i*) first, the linearized, distortions-sensitive transformation of $P(t)$ data is carried out; it indicates domains where the validated application of a given scaling equation is possible and yields optimal values of relevant parameters, (*ii*) the final fitting via the selected scaling equation is carried out for identified domains, and using obtained optimal values of parameters. The analysis reveals links between $G_P(P)$ local 'disturbations' and some historical and prehistorical reference events, showing their global scale impacts.

## Introduction

The Anthropocene epoch began 12,000 years ago, only six millennia after the last Ice Age started to end. About 2–4 million people lived on Earth then [1]. Almost twelve millennia later, in 1800, the global population reached 1 billion [2,3]. It took 125 years to add the next billion to the World's population. In November 2011, the global population was 7 billion, and only 11 years later, 8 billion [4].

**Data availability statement:** All relevant data are given in the Appendix, submitted within the report.

**Funding:** National Center for Science (NCN, Poland), grant ref. 2022/45/B/ST5/04. The funder had no role in study design, data collection and analysis, decision to publish, or preparation of the manuscript.

**Competing interests:** No competing inererests between authors.

In the 21st century mobile phones and online information exchange systems, supported by artificial intelligence, are omnipresent. The industries based on global supply chains are the norm. Once the ongoing process of developing and implementing hypersonic transport terminates, travels between the most distant places on Earth will be reduced to a few hours. 'Globalization', referring to interactive human populations in the spatially constrained system of the Earth, is becoming a fact. The emerging '*Brave New World*' [5] is threatened by the collapse of the social and political order, if not the civilization. One can recall fast-spreading pandemics, the enormous Climate & Global Warming and Energy Crises, migration waves, and wars matched with political disorders. The latter is often associated with global-scale targets of dictators, predatory states, and organizations.

It might seem that today's times, driven by extraordinary technological innovations and grand problems and challenges, are exceptional. However, people living in England or Scotland at the beginning of the 19th century, when the 1st Industrial Revolution was becoming omnipresent, could have had similar feelings. The Steam Age innovations were quickly and widely implemented, yielding previously unimaginable technological achievements but also leading to political and socio-economic turbulences. Rapidly growing, industry-driven cities were overcrowded and noisy, with choking smoke and dramatically polluted rivers [6,7]. In the 21st century, the times of 4th and 5th Industrial Revolutions [8,9], challenges and problems are similar but at a truly global level.

Consequently, viewing past population trends and forecasting future changes are essential for global insight, planning, and governance. Various national and international agencies and independent researchers focus on modeling global population changes. Nevertheless, the problem remains puzzling, as shown by the fan of global population forecasts ranging between 6.3 and 14.5 billion, even for the relatively close period 2050–2100 [10–13].

There are two leading cognitive paths for modeling global population changes.

The 1st path focuses on scaling equations describing long-range population changes, which can validate nearest future extrapolations. It was initiated by the pioneering works of Malthus (1798) [14] and Verhulst (1838) [15]. The latter directly introduced the factor describing the impact of available resources on population changes. Since then, many other scaling equations for modeling global population growth have appeared [16–35]. However, the Malthus and the Verhulst models have remained a significant reference [36–53].

The 2nd cognitive path aims to define the global population, considered via reference impacts of geographical regions, social groups, changes in education, multiple aspects of social interactions - especially regarding the role of women, migration issues, education, birth/death ratio, age structure, economic development... Such multitude of data are analyzed statistically in frames of models developed in management and econometrics, bio-evolution, or socio-economic sciences [24,33,54–65], which have shown their effectiveness for various problems from the scale of states to companies and corporations, and also for multiple issues in biology, ecology, medicine, … [24,54,57,61,62]. For this path, links between mentioned factors, often in feedback interactions, are essential. It has

to be supported by weightings based on expert opinions, raising the question of subjective arbitrariness and reliable error estimations. For this path, the direct application of autoregressive-moving-average (ARMA) or ARIMA (autoregressive integrated moving average) [66–68] might seem a workable solution. It is related to the statistical analysis of processes developing in time series using autoregression and moving averages, often using polynomials (second or first order) as the reference tool [67]. They are broadly applied to discuss the time-related changes of different properties in econometrics [68,69] or medicine-related issues [69–72]. They can also be implemented for population studies, both time-related portrayal and forecasting, to avoid the knowledge of an underlying scaling equation. Such an approach describes the population and related issues and the development of urban centers, regions, or countries [73–80]. Generally, the recurrent approach underlies the vast majority of analysis within the mentioned 2$^{nd}$ cognitive path for global population $P(t)$ studies.

Notwithstanding, the canonic ARMA/ARIMA modeling is hardly used for global $P(t)$ modeling [70–83]. It can be explained by the fact that they require multi-dimensional and high-accuracy data, preferably for the same (minimal) time steps, which for the global population ceases to be available when shifting to past times. For forecasting, the cumulated error of parameters is significant, which can lead to discrepancies reaching even 30% for only 2–3 decades of extrapolations [82,83]. Notably, these methods offer data portrayal but weakly address the nature of underlying processes. Finally, it is worth indicating that recent distortions-sensitive analysis of the global population growth revealed the significance of non-monotonic and aperiodic events [84], which is inherently beyond the ARMA/ARIMA approach.

The primary inspiration for this report was the recent paper by Lehman et al. [62], which combines the mentioned basic 1$^{st}$ and 2$^{nd}$ cognitive paths for global population studies and considers it in frames of the Verhulst-type scaling equation associated with the extending concept by Pearl and Reed [85,86], further developed by Volterra [87] and Cohen [88]. We stress this issue because such an approach essentially extends the basic Verhulst (Two-Mode Logistic (TML) or bimodal) approach, often questioned for its suitability for predictive purposes regarding human populations. Nevertheless, the Verhulst equation remains a significant reference for developing 21$^{st}$ century Sustainable Civilization matched with the Circular Economy [89–91]. The carrying capacity (resources) factor is often correlated with ecological constraints, such as Global Warming, environmental pollution, the grand energy crisis, or crucial raw materials shortage.

In ref. [62] by Lehman et al., the plot of per capita relative global growth (RGR) of the population $G_P^i = \left(\frac{1}{P_i}\right)\left(\frac{\Delta P_i}{\Delta t_i}\right) = \left[\frac{\left(\frac{\Delta P_i}{P_i}\right)}{\Delta t_i}\right]$ vs. $P_i$, where the latter means the population for selected subsequent times in the range $10,000\ BC < t < 2010$, $\Delta P_i$ is for population steps in subsequent time periods $\Delta t_i$, is considered. The plot revealed explicit linear patterns of $G_P(P)$ changes: from $\sim 10,000 BC$ to $\sim 1962$ and subsequently from $\sim 1962$ up to $2010$, with qualitatively different slopes and the crossover at $P_{cross} = 3 - 3.5 billion$. The plot $G_P(P)$ vs. $P$ was used as the argument for portraying global population changes via the Verhulst-type equation with the sequence of the growth rate ($r_i$) and carrying capacity ('available resources': $s_i$) coefficients.

This report focuses on the meaning of this exceptional (universalistic?) pattern of $G_P$ for global population changes [62]. The analysis explores the new generation of global population data obtained via the numerical filtering of inherently scattered data from different sources. It enabled the discussion of the analytic counterpart for $G_P(P)$ changes. The distortions-sensitive insight revealed local disturbances in global population changes, correlating with some socio-economic and historical events. The report also presents new conclusions regarding global population changes Verhulst-type scaling.

## Remarks on Malthus and Verhulst equations

The turn of the 18$^{th}$ and 19$^{th}$ centuries was associated with the rising wave of the 1$^{st}$ Industrial Revolution. Rapidly growing industrial centers explored breakthrough technological innovations of the Steam Age [6]. Developing industry-driven urban centers were overcrowded and full of hope for a new life, but there was also enormous poverty and social unrest [6,7]. In those times, the Scientific Method [92,93] had already become a leading cognitive method that supported the innovations-driven Industrial Revolution. This was primarily due to Isaac Newton's legacy, which ranged from physics and mathematics to economics [92]. Newton

showed the ultimate importance of empirical verification and adequate descriptions of the laws of nature using functional scaling relations, including the differential analysis he introduced. The unified description of the motion of an apple falling from a tree and planets or comets 'in the sky" remains a crucial example of Newton's grand universalistic success [92].

These inspirations declared Robert Malthus, who formulated the first and still significant model scaling for describing population changes $P(t)$ [14,37,40]:

$$\frac{dP(t)}{dt} = rP(t) \quad \Rightarrow \quad P(t) = P_0 e^{rt} \quad \Rightarrow \quad lnP(t) = lnP_0 + rt \tag{1a}$$

$$G_P(P) = \frac{1}{P(t)}\frac{dP(t)}{dt} = \frac{dlnP(t)}{dt} = r \tag{1b}$$

where time $t$ refers to the onset time $t_0$, and it is matched to the prefactor $P_0$; the Malthus growth rate coefficient.

$$r = const$$

The left part of equation (1a) is for the basic differential equations illustrating the Malthus model, the mid part is for the Malthus equation, and the right one shows the linear behavior of $P(t)$ changes in the semi-log scale.

Equation (1b) presents the Malthus model in terms of the per capita relative growth rate (RGR, $G_P$), which is the focus of the given report. Malthus recognized the meaning of resources (food) amount considered via a separate equation that assumed their much weaker, linear growth: $F(t) = a + bt$. Malthus commented on the hypothetical feedback of the population and food changes [14]: '*The population increases in geometrical ratio and the subsistence rises only linearly, which finally leads to times of 'vice and misery*'. It is the famous Malthusian Trap (Catastrophe).

In 1838, Pierre François Verhulst introduced for studying human population changes the model where the impact of resources (food) is included in the scaling equation [15,19–23]:

$$\frac{dP(t)}{dt} = rP - sP^2 \quad \Rightarrow \quad K = \frac{r}{s} \quad \Rightarrow \quad \frac{dP(t)}{dt} = rP\left(1 - \frac{P}{K}\right) = rP\left(\frac{K-P}{K}\right) \tag{2a}$$

$$\Rightarrow \quad G_P = \frac{1}{P}\frac{dP(t)}{dt} = \frac{dln(t)}{dt} = r - sP(t) = r - r\left(\frac{P(t)}{K}\right) \tag{2b}$$

The left part of Eq. (2a) is for the reference Verhulst model differential equation, $r$ denotes the Malthus growth rate, and $s$ describes available resources (originally food): essentially $r, s > 0$, and $r, s = const$. Pearl and Reed [85,86] popularized the version of the Verhulst model reference equation with the carrying capacity $K = r/s$ factor, shown in the right-hand part of Eq. (2a). The carrying capacity $K$ can be considered as the maximal, 'equilibrated' population that can stay in a given system with existing resource constraints. It is associated with the 'stationary phase of the describing Verhulst bimodal function $P(t)$, namely: $K = limP(t)$ for $(t \to \infty)$.

Equation (2b) presents the basic Verhulst model differential equation in frames of the per capita relative growth rate (RGR, $G_P(P)$), showing its linear behavior.

Notably, that already in 1760 Danielle Bernoulli considered the Verhulst relation counterpart for testing the mortality caused by smallpox. Implementing Bernoulli's analytic path, one can derive the Verhulst: model relation for $P(t)$ changes [20]:

$$\frac{dP(t)}{dt} = rP - sP^2 \quad \Rightarrow \quad \frac{1}{P^2}\frac{dP}{dt} = \frac{r}{P} - s \quad \Rightarrow \quad p = \frac{1}{P} \quad \Rightarrow \quad \frac{dp}{dt} = \frac{r}{K} - rp = -r\left(p - \frac{1}{K}\right) \quad \Rightarrow$$

$$\Rightarrow \quad q = p - \frac{1}{K} \quad \Rightarrow \quad \frac{dp}{dt} = \frac{dq}{dt} = -rq \quad \Rightarrow \quad q(t) = q_0 \exp(-rt) \quad \Rightarrow$$

$$\Rightarrow \quad \frac{1}{P} - \frac{1}{K} = \left( \frac{1}{P_0} - \frac{1}{K} \right) \exp(-rt) \quad \Rightarrow \quad P(t) = \frac{K}{1 + CKexp(-rt)} \tag{3}$$

where $C = \frac{1}{P_0} - \left( \frac{1}{K} \right)$; for $(K \to \infty) \Rightarrow P(t) = P_0 \exp(rt)$, i.e., it reduces to the basic Malthus Eq. (1).

For isolated systems with a constant amount of resources (food), despite the rising population the above Verhulst relation describes the bimodal behavior, starting from the Malthus-types (Eq. (1)) rising 'phase' and terminating with the stationary 'phase' where $P(t) \to K$ for $t \to \infty$ [15,28–30,85–89]]. Such behavior occurs for systems with renewable resources, where $s, K = const$. Notable, that the of $G_P(P)$ can validate the Verhulst model description via the emergence of the linear behavior, indicated in Eq. (2b). Subsequently, the linear regression yields optimal values of $r, s, K$ parameters with reliable error estimations. These values can be substituted to Verhulst Eq. (3) for $P(t)$ data portrayal. Thus, nonlinear fitting, which is always associated with a significant error in derived parameters, can be avoided.

Such a protocol for data treatment recalls the derivative-based and distortions-sensitive analysis introduced by one of the authors (A. Drozd-Rzoska) for studying the properties of soft matter complex systems [94–101].

For isolated systems with non-renewable resources that are continually and irreversibly consumed by a growing population, the stationary phase is relatively short, and followed by the population decline due to exhaustion of resources. Such a picture occurs in microbiological tests for populations of bacteria or yeast in a container isolated from the surroundings and a given and non-replenished amount of food (like sugar) [45,47,48,50,53]. As for more complex systems, it is worth recalling the model developed by Tilman [102,103] and followers [104], which discussed resources interacting with population growth, which indicates carrying capacities determined by resource needs.

The Verhulst-type pattern has recently been shown for human population changes on Easter Island (Rapa Nui), the Pacific island, located well remote from other islands and the South American mainland [105]. Although it fairly portrays population data, it is worth mentioning that recent studies have shown that the previously dominant picture related to isolation, limited resources, and ecological constraints should be changed. Recent research communications have shown the devastating impact of contact with European sailors and later marauders who enslaved people and kidnapped them to the South American mainland [106,107]. Nevertheless, the 'idealistic pattern' discussed for Rapa Nui describes population changes in industrial cities created by a dominant industry [105]. It is the case of Detroit (IL, USA), associated with the automobile industry, and Bytom (Silesia, Poland), a former coal mining center [105], for instance.

The basic Malthus and Verhulst scaling relation remains a significant reference tool for modeling population changes from microbiology [108,109] and food technology [110,111] to the spread of epidemic outbreaks [48] growth of some animals and plant populations [50] to some problems in economy and management [39,40,51], and physics for nonlinear dynamical systems in the presence of random perturbations[112–114], which seems to fairly correlate with extremely complex global population. Nevertheless, the explicit validity of Malthus and Verthulst equations for the global population changes remains a challenge [23,84].

One can also consider a third, hardly discussed, option of population changes resulting from the Verhulst scaling relations, especially for isolated (closed) systems with limited resources and carrying capacity. For such systems, a relative increase of resources due to a reduction in population requirements/needs can occur. In the language of physics, it can be considered a spontaneous self-adaptation of complex active matter population to the system's constraints [114]. To illustrate this route (3rd path), one can recall the case of pygmy mammoths [115,116]. Near 10,000 BCE, rising ocean levels cut off mammoths on Channel Island, the west coast of North America. The last of them lived only 4,000 years ago. The evolution caused their height to be only $1.7 - 2\ m$, and their weight was even 10x less than for original Columbian mammoths [115,116]. Such a reduction led to a new equilibrium, increasing the number of available resources and space and allowing for prolonged survival. The final disappearance of the pygmy mammoth is linked to genetic degenerations, i.e., 'internal' population problems [115,116].

For the global human population developing within the Earth's spatial, resources, and ecological 'constrained' capacities, the 3rd path can mean a sustainable civilization with rational energy consumption and minimal environmental harm. Such a civilization pattern can reduce global-scale threats in the 21st century [117].

The question arises of whether such a 'sustainable society' strategy has already appeared in the past. For the authors, the origins of Slavic tribes in the early Middle Ages are worth considering here. Pre-Slavic tribes appeared in Central Europe 'suddenly' between 5th and 7th centuries CE. It was a time of climatic breakdown, the peak of which was the so-called emperor Justinian winter, associated with the temperature in Europe, and perhaps globally, dropping by as much as 1–2 K average per year. In Central Europe, winters became long and extremely cold [118]. It led to essential vegetation and crop problems for farming communities. Such conditions were one of the motivations for the great migrations of Germanic tribes from Central Europe to the Roman Empire, located in a more favorable climate. Finally, it led to the fall of the Western Roman Empire and the long-term problems of its eastern part, linked to Constantinople [119]. Suddenly, in Central Europe's 'abandoned' areas, traces of small communities with a surprisingly 'primitive' way of life appeared. They are associated with pre-Slavic tribes, whose original habitats are often linked to unspecified locations in 'deep' Eastern Europe [120,121]. However, recent genetic research has shown that the ancestors of the proto-Slavics lived in central Europe at least 500 years before the mentioned times [122], probably peacefully coexisting with Germanic tribes. During the 'climate catastrophe' times, Germanic tribes chose migration to solve the problem, which led to the conquest of the Western Roman Empire. A part of the population, closely related to agricultural life, seems to have remained in Central Europe. Dugouts in which they lived are often indicated as the hallmark of their 'primitivism' [120–122]. However, such shelters are also the most effective way to survive under extreme conditions. This situation can also be seen as transitioning to a 'sustainable society', adapted to the climate crisis conditions. It could also be a significant formative period for Slavic tribes and the source of their enormous success in the 8th and 9th centuries [119–122], as the climate warmed and available resources increased.

In the Anthropocene period, a continuous global population growth occurs. It is related to nonlinear changes in the semi-log plot $lnP(t)$ or $\log_{10} P(t)$ vs. $t$ [123]. Such behavior is beyond the basic Malthus pattern (Eq. (1)), which can be named the Super-Malthus behaviors, following the name proposed in ref. [84]. Such behavior is also beyond the basic Verhulst behavior described above.

However, a century ago, Pearl and Reed [85,86] suggested that human population growth may follow a sequence of Verhulst scaling equations coupled with a sequence of carrying capacities for which the transition occurs well before the previous one terminate is approached. Consequently, the population growth pattern may pass through successive Verhulst-type steps without any distinctive manifestation of the Verhulst plateau [85,86]. Such an analysis made it possible to describe the population changes in the USA up to 1930 [85,86]. In the subsequent decades, the growth of the US population was significantly greater, since it is an open system, contrary to the global population. In 1928, Volterra [87] developed the concept of barriers crossovers', focusing on animal species living together as an example. Cohen implemented it for the global human population growth model description (1995, [88]).

Recently, Lehman et al. [62] have developed these concepts by considering global population growth in terms of three successive bio-ecological levels: (1) interactions with predators, (2) interactions with prey, and (3) intraspecific interactions. Global population changes at each level are governed by level-dependent ecological coefficients ($r_i$; $s_i$), $i = 1; 2; 3$. These led to population discontinuities progressively separating (*i*) a primordial phase, where pre-human ancestors interacted with their environment as other animals do, (*ii*) a mastery of tools, fire, and specialization phase, (*iii*) an agricultural phase, and finally (*iv*) a present controlled-fertility phase. Parameters for population growth changed at each discontinuity. The basic justification for such behavior was linear changes in the per capita of the population relative growth rate (RGR) $G_P(P) = \left(\frac{1}{P}\right)\left(\frac{\Delta P}{\Delta t}\right)$ plotted against the population itself, with different signs of slopes related to $s$ parameter [62]. These were implemented for the following discrete Verhulst-type equations [62]:

$$G_P = \frac{1}{P(t)} \frac{\Delta P(t)}{\Delta t} = r_i \pm s_i P(t)$$

(4)

where coefficients $r_i, s_i = const$ are for subsequent time domains differently subjected to time-varying bio-/eco- factors.

In the above relation, the sign '$\pm$' reflects the occurrence of both $s_i > 0$ and $s_i < 0$ and just such behavior was evidenced for $G_P(P)$ changes in Fig 3 of ref. [62]: (*1*) the linear domain for the period lasting almost 12 millennia, $10,000\ BCE < t < 1962 \pm 5$, where $r_1 > 0$, $s_1 > 0$, and (2) for the period $\sim 1962 < t < 2010$, related to $r_2 > 0$, $s_2 < 0$. Using the mentioned results [62], one can estimate the crossover between these domains at $P(t_{cross}) \approx 3.4 \pm 0.2$ *billion*. The final analysis used 98 global population data covering nearly 12 millennia [62].

The second domain (*2*), starting near the mid of sixties in 20$^{th}$ century, satisfies the conditions for the standard Verhulst model behavior, defined via Eq. (2b) above.

For the first domain (*a*), lasting ~12 millennia, the RGR factor follows the anomalous pattern: $G_P = r + sP$. Recalling the Verhulst model reference Eq. (2b), it is related to the carrying capacity $K < 0$, which is a puzzling result in frames of this factor meaning discussed above. Moreover the mentioned linear behavior is associated with a single pair of $(r_1, s_1)$ parameters, but their substitution to the Verhulst Eq. (3) does not portray $P(t)$ data. The description can be reached using a set of $(r_i, s_i)$ parameters.

Notwithstanding, Fig 3 in ref. [62] shows a unique 'empirical' universalistic pattern for the global population growth from the Anthropocene onset till 2023.

One can consider two cognitive paths to comment/explain this unique finding.

First, one can focus on $G_P(P)$ changes concerning two apparent Malthus-type growth rates $r'$ and $r''$, namely:

- For the standard Verhulst-type pattern in the domain (*2*): $G_P = r' = r - sP(t) = r - r\left(\frac{P(t)}{K}\right)$. The apparent growth rate continuously decreases $r' \approx r \rightarrow r' \approx 0$ reflects the bimodal behavior, i.e., from the near-Malthus to the stationary behavior.

- For the anomalous behavior in the domain (*1*): $G_P = r'' = r + r\left(\frac{P(t)}{K}\right)$. The apparent growth rate increases with the rising population: the rising population seems to increase the system's carrying capacity (Earth) continuously.

More significant insight can be reached by recalling the model analysis by Cohen [88], who considered the basic Verhulst model relation (Eq. (2a)): $\frac{dP}{dt} = rP\left[\frac{(K-P)}{K}\right]$ in frames of the Enlightenment epoch philosopher Marquise Jean-Antoine-Nicolas de Condorcet expectations that the 'human mind' is capable of removing all obstacles to human progress [124]. For the problem considered here, people can permanently expand Earth's carrying capacity, including the extraordinary rise in food production. In the Industrial Revolutions epoch, novel methods in agriculture have increased crops despite the relative reduction in cultivated areas. Innovative food preservation methods qualitatively reduce microbiological threats and food losses in the lengthening logistics chain [125]. Cohen posited the following relationship between changes in global population and the carrying capacity [88]:

$$\frac{dP(t)}{dt} = c\frac{dK(t)}{dt}$$

(5)

where '*c*' was named the Condorcet parameter [88].

For $c = 1$ each additional person contributes to the carrying capacity as much as they consume, which leads to exponential population growth described by Malthus' relation: (Eq. (1)). For $0 \leq c < 1$ each additional person influence available, near constant, carrying capacity. The per capita consumption reduces with the passing of time until reaching the stationary state. It is related to the standard, bimodal (logistic) Verhulst behavior. The condition $c < 0$ leads to a diminishing population. For $c > 1$, each additional person yields a significant carrying capacity added value above their own needs and wants [88]. It leads to the super-Malthusian [84] rise of the global population [84], matched to the anomalous behavior of $G_P(P)$ in the first millennia [62]. Cohen showed that the case $c > 1$ could explain even the extraordinary population growth via the 'hyperbolic' Doomsday relation, suggested for the period ~400CE till 1958 by von Foerster et al. [16]. The implementation of Cohen's reasoning for the Pearl and Reed concept extending Verhulst modeling, developed further in ref. [62], can

conceptually explain the transformation from Malthus to Super-Malthus [84] growth occurring for the global population. It also shows the possible significance of the Condorcet parameter, particularly for the carrying capacity concept.

## Materials and methods

This report explores the new way of data preparation based on collecting global population data from various sources and their numerical filtering using the protocol introduced by one of the authors in material engineering and glass transition physics studies [84,98–101]. It enables finding optimal evolution paths in a set of inherently scattered 'noise-like' data via employing the Savitzky-Golay filtering principle with the support of Origin and Mathematica software. The Savitzky–Golay method is a smoothing numerical filtering procedure that can be used to reduce 'noisy' distortion of digital data, i.e., to increase their precision without distorting the signal tendency [126,127]. In the given report, 'empirical' data from refs. [128–134] were prepared in such a way. Finally, a 'smooth' set of 193 population data from 10,000 BCE to 2023 has been obtained. Such a way of data preparation enabled the linearized distortions-sensitive and derivative-based analysis [84], for which emerging linear domains indicate the periods for which the selected scaling equation can be applied to portray $P(t)$ changes. Applying the standard linear regression protocol yields optimal values of relevant parameters with well-defined errors [84,98–101]. It should be noted that global population data are always burdened with estimation error, increasing with the distance from modern times. The estimates significantly depend on ongoing historical, archaeological, or genetic research for previous historical epochs. It means that global population data must be permanently updated, and earlier estimates should be critically considered. The global population data obtained following the above protocol are given in the S1 Appendix.

## Results and discussion

Fig 1 shows global population changes from Anthropocene (10,000 BCE) onset to 2023, based on data prepared via the protocol recalled above. The inset in Fig 1 focuses on the ongoing Industrial Revolutions [7–9] times. The arrows indicate

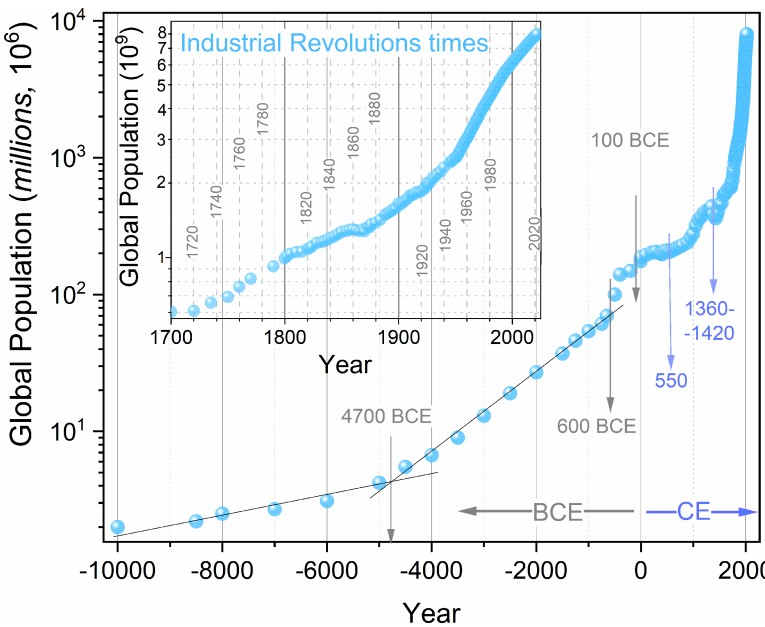

**Fig 1. The plot showing global population P(t) changes, in a semi-logarithmic scale, from 10,000 BCE to 2023. It is based on the data given in the** S1 Appendix. **The inset focuses on the Industrial Revolutions [6–8] epoch. The arrows indicate some characteristic dates/periods manifested in the plot.**

emerging characteristic changes in the evolution of the global population. For almost 10,000 years, up to ~600 BCE, which can be correlated with the definitive end of the Bronze Age or the development of great civilizations in the Mediterranean area and China [135,136], global population changes can be portrayed by the basic Malthus relation (Eq. 1), as shown by the linear behavior in the semi-log plot. However, there is a significant change in the slope of such Malthusian behavior around 4700 BCE, which may be related to the acceleration of population growth: the Malthus rate coefficient increased $4.6\times$ after the year 4700 BCE. Between 100 BCE and 500 CE, a plateau in global population changes appears. It remains constant at 190–200 million global population level. This period correlates with the Roman Empire times [136–139]. Its population reached 40 million, but even 70 million has recently been indicated at its peak development times [137,138]. The Empire could include between $\frac{1}{5}$ to even $\frac{1}{3}$ of the global population. The enormous success and the fall of the Roman Empire have remained the subject of research and fascination for generations of historians [135–139].

We want to draw attention to a factor important for the population discussed in frames of the Verhulst model: the available/necessary resources or carrying capacity. In Roman Empire times, slavery was a 'social norm'. However, enslaved people had an additional meaning in the Empire; they were also the crucial 'energy resource' that drove the economic system, explored at the extreme 'global' scale. The enslaved built omnipresent imperial buildings, aqueducts, roads, channels, and tunnels that remain symbols of the Roman Empire's epoch. They were also essential for the 'industry'. For instance, there were giant silver mines in Rio Tinto (Iberia), and between 20 and 50 thousand enslaved people worked there [138]. The great historian Pliny (*Gaius Plinius Secundus*) remarked that each could survive between 6 months and 2 years [138,140]. Using modern language, for Roman managers, enslaved people were a kind of an 'energy resource' and permanent 'new supplies' well required in the 'business plans'. Terrifying. Wars and expeditions into 'barbarian' territories to gain slaves ('human energy') were necessary for the high level of the Imperial economy. However, the Empire weakened, and new 'human energy supplies' diminished. According to Verhulst's model, a lack of significant resources has led to population decline.

Fig 1 also shows the strong impact of the Black Death epidemic that devastated Asia and Europe in the 14th and 15th centuries, leading to a catastrophic decrease in World population [141,142].

When discussing the global population and its relation to the Verhulst-type scaling, one should consider the extension of Eq. (4) for per capita population growth $G_P$ to the case of 'smooth' population data, where the derivative analysis is possible:

$$G_P(P) = \lim \left[ \frac{1}{P(t)} \frac{\Delta P}{\Delta t} \right]^{\Delta P \to 0}_{\Delta t \to 0} \Rightarrow G_P(P) = \frac{\frac{dP(t)}{P(t)}}{dt} = \frac{d\ln P(t)}{dt} \tag{6}$$

The above analytic definition requires a new definition of time $t$, which is irrelevant to the standard 'discrete' definition (Eq. (6)). In this report, the time scale is considered since the Holocene 'harbinger', estimated at 12,000 BCE. It is $\sim 4,000$ years after the last grand glaciation (Ice Age) ended, and since then, global temperatures have risen by $\sim 4^oC$ [1]. The great ice sheets had receded from Europe, but sea levels were still lower than today. It meant, for example, the existence of the Doggerland, a large landmass in what is now the North Sea, i.e., nowadays submerged [143,144]. All of Europe, including Scandinavia and today's British Isles, was opened for wandering Homo Sapiens.

Fig 2 shows the results of the derivative analysis for the global population data shown in Fig 1, in frames of the RGR factor $G_P = \frac{d\ln P}{dt}$ The obtained picture agrees with the results presented in Fig 3 of ref. [62] by Lehman et al., where the standard, discrete definition of $G_P$ (Eq. 6: left part) was used. In ref. [62] In Fig 2, two linear domains appear, with a crossover in the mid-sixties. As discussed above, they are related to the 'standard' and 'anomalous'. Parameters describing these lines, in reference to Eq. (2b) are given in Table I. Fig 2 contains the extension of per capita relative population changes up to $G_P(P) \to 0$, linked to $P_{\max}$. It can be related to reaching the hypothetical stationary 'phase' following the above discussion regarding the Verhulst function features. The usage of the new set of $P(t)$ data, supported by the numerical filtering, also reveals that the second-degree polynomial offers a better representation of the changes in $G_P(P)$ when considering the multi-millennial period from the Anthropocene onset. In fact, explicitly linear behavior seems to be reliable

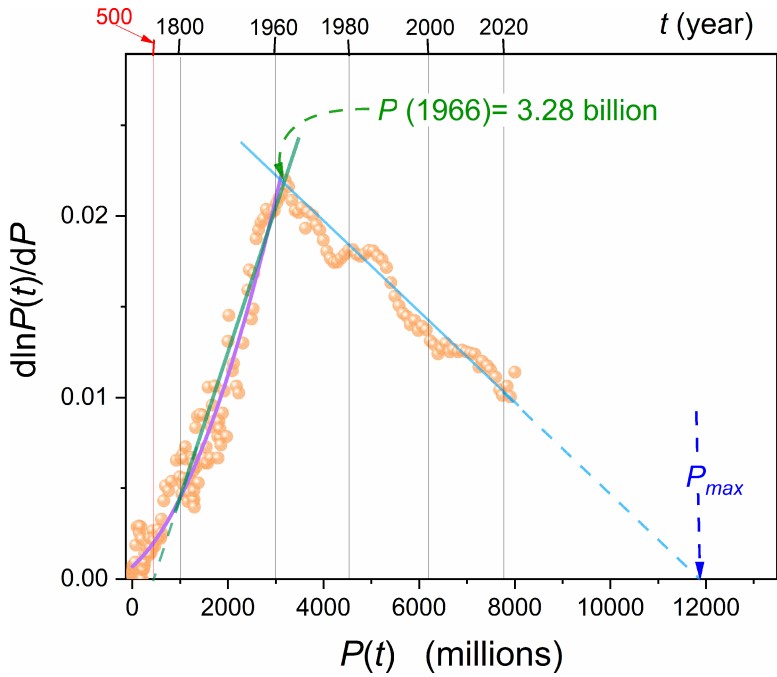

**Fig 2. Changes of the per capita relative world population growth $G_P(P)$ determined by the derivative analysis defined by** [Eq. (6)] **and based on data shown in Fig 1 and collected in the** **S1 Appendix**. **The crossover between the two emerging domains is shown. The extrapolation to $G_P(P_{max}) = 0$ indicates the onset of the stationary phase, which can be associated with the maximal population. Note the 'squeeze/compression' of the first 10 millennia of global population growth caused by the scale applied. Table I and its caption give parameters related to linear domains (in green and blue) and for the polynomial portrayal (in violet).**

only since ~1800. The linear and polynomial parameterization explicitly overlap only from ~1950. In [Fig 3] of ref. [62], which parallel [Fig 2] of the given report, the linear domain portrayal was used from 10 000 BC to ~1962. Following Eqs. (2) and [(3)] it is related to single pairs of $(r, s)$ parameters, each coupled to a single Verhulst relation ([Eq. (3)]). However, the substitution of these value does not yield any $P(t)$ portrayal. In ref. [62], the portrayal was reached using a sequence of Verhulst equations with different values of $(r, s)$ parameters. Hence, a formal inconsistency appears. To comment on this issue, worth noting is the fact that a better portrayal of $G_P(P)$ data in [Fig 2] for the mentioned extreme multi-millennial period, a second-order polynomial offers a better portrayal. It is shown by the violet curve, with parameters in the caption of Table I. Such nonlinear changes of $G_P(P)$ can justify the multi-functional portrayal applied in ref. [62] for $P(t)$ changes.

The consequence of the huge change in the magnitude of $P(t)$ and the time scale values for the data presented in [Fig 2] has to yield data 'compressing' and superposition for a colossal time period, covering more than 10 millennia. This problem can be avoided when presenting data in the log-log scale, as in [Fig 3]. It reveals that the explicit linear pattern in $G_P(P)$ changes occurred only after $\sim 700AD$, and continuous until the crossover at $\sim 1966 - 1970$. It seems that this trend began at the time of the King and Emperor Charles the Great, Charlemagne, nowadays considered the modern Europe 'father' [145]. The pattern was definitively different for the earlier multi-millennial periods, with explicit correlations to characteristic historical epochs, as shown in [Fig 3].

Following Eq. (1), one obtains for the basic Malthus Eq. (1): $G_P = r = const$. [Fig 3] shows that such behavior explicitly occurs only in the late Neolithic period and times of 'classic' ancient empires in Persia, Greece, or Macedonia between $800BC - 100BC$. The analysis concluded in [Figs 2] and [3] can be considered a subtle, distortion-sensitive validation tool for scaling relations describing global population changes.

Table I presents relevant parameters describing the mentioned linear domains for $G_P(P)$ expressed by subsequent growth rate $r$ and carrying capacity $s$ parameters (Eq. 4). It suggests that the pre-crossover domain related to times between $\sim 700\ CE$ to $\sim 1968 \pm 5$ and population $P < 3.3\ billion$ should be described by a single Verhulst equation with parameters given in Table I. The same can be expected for the post-crossover domain, which has been extended till today. Nevertheless, substituting these parameters to the Verhulst equation does not lead to $P(t)$ portrayals in the mentioned domains. Consequently, a question if the Verhulst model scaling is appropriate for describing global population changes arises. Regarding the crossover year (1966), the error related to three standard deviations and the intersection of two lines is notable in Fig 3.

Table 1 Values of the parameter characterizing the linear domains for the per capita population growth rate $G_P(P)$, defined by Eqs. (2 and 6), and shown in Fig 2: $G_P(P) = r - s \times P$, for domains indicated in the Table. The fitting results are related to the linear regression standard procedure. Note that substituting these parameters to the Verhulst equation in indicated time domains does not reproduce $P(t)$ changes.

The polynomial in Fig 2 is related to $\frac{dlnP(t)}{dt=}6.81 \times 10^{-4} + 2.31 \times 10^{-6}P + 1.48 \times 10^{-9}P^2$: it coincides with the linear approximation since 1950 (population ~2.5 billion).

| time period | population range | intercept $r$ parameter | slope $s$ parameter |
|---|---|---|---|
| 1st domain 700 CE - 1966 | 1 million – – 3.3 billion | $\begin{pmatrix} -0.46 \\ \pm 0.07 \end{pmatrix} \times 10^{-2}$ | $\begin{pmatrix} 8.1 \\ \pm 0.1 \end{pmatrix} \times 10^{-6}$ |
| 2nd domain 1966 - 2023 | 3.3 billion – – 8.1 billion | $\begin{pmatrix} 2.81 \\ \pm 0.1 \end{pmatrix} \times 10^{-2}$ | $\begin{pmatrix} -2.27 \\ \pm 0.05 \end{pmatrix} \times 10^{-6}$ |

The authors want to stress the 'reversed' analytic route compared to the standard pattern applied so far. The standard analysis is related to fitting $P(t)$ data using a selected scaling model equation in a subjectively chosen time domain. In this report and the related very recent report of the authors [84], 'empirical' $P(t)$ data are first directly tested via the linearized derivative-based transformations (Eqs. 4 and 5). It indicates domains where the given equation can describe 'empirical' data, also delivering optimal values of basic parameters. The final fitting of $P(t)$ data is reduced solely to the prefactor. Such protocol succeeded in ref. [84] for $P(t)$ portrayals via super-Malthus equations and numerous studies in critical and glass-forming physical systems [98–101]. However, such a procedure failed for the analysis recalling the behavior shown in Figs (2) and (3) in frames of Verhulst equation.

Fig 4 presents changes in the global population $P(t)$ in contemporary times, since ~1940 till nowadays, including the crossover at $P(1966) \sim 3\ billion$. The shows the behavior of $G_P(P)$ since the crossover till 2023, supplementing results presented in Fig 3. It confirms the linear pattern of changes, with local disturbances coinciding with some global scale events: (*i*) 1973 can be associated with the contestation of the existing social order by the young generation, which influenced its changes; it is also the first energy crisis (oil crisis), (*ii*) the decade of the eighties is the final stage of the Cold War and political and economic changes that the presidency of Ronald Reagan can embody, (*iii*) the end of the Cold War and the fall of communism is the year 1990; a year or two later, a group of new countries joins the free-market World, (*iv*) the next characteristic date is 2009, i.e., the beginning of the great global banking and economic crisis, (*v*) 2018 is the time of the COVID19 pandemic crisis.

The behavior shown in Fig 3 and the inset in Fig 4 is related to the semi-log scale presentation. These results can suggest the preference for an exponential–type description Super-Malthus behaviour of per capita relative population growth changes, namely:

$$G_P(P) = \frac{dlnP(dt)}{dt} = a \times \exp(bP) \Rightarrow G_P(P) \approx a + (ab)P + \ldots \tag{7}$$

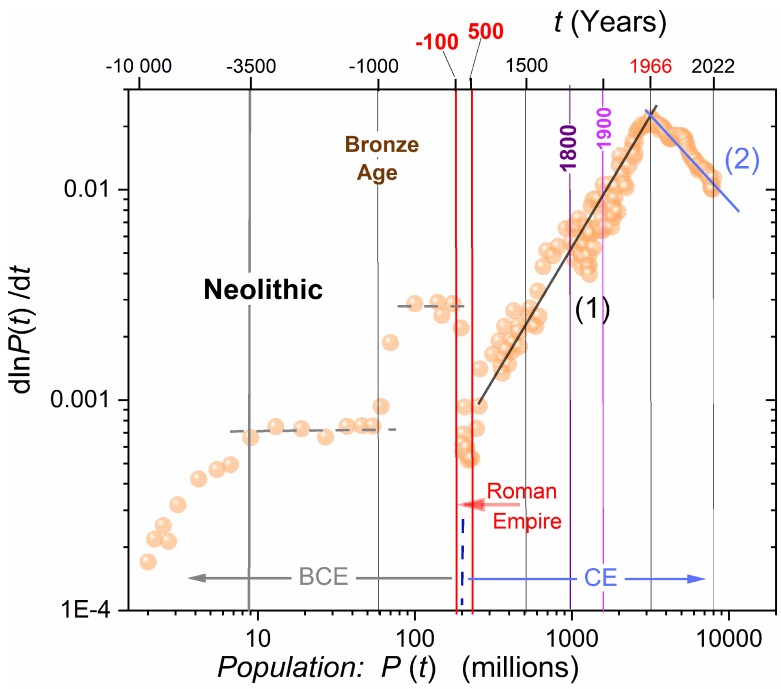

**Fig 3. The log-log scale presentation of the per capita growth of the global population $G_P(P)$ data, shown in the linear scale in Fig 2.** Emerging relevant historical domains are indicated. It is visible that the hypothetical 1st linear domain visible in Fig 2 can be considered only from early Medieval times.

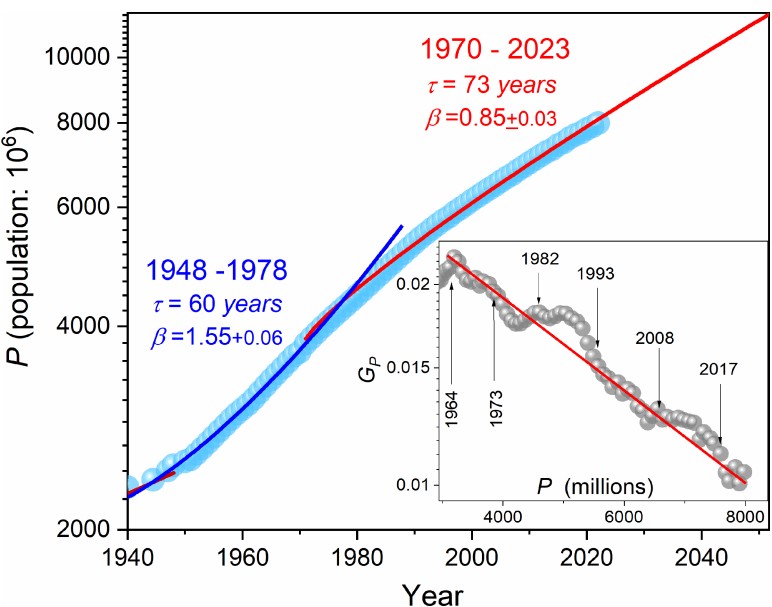

**Fig 4. Changes in the global population from ~1940 till 2023.** The parameterization is related to the empowered exponential Super-Malthus Eq. (10) [84], with the parameters given in the plot. The inset recalls data from Fig 2 but is presented in a semi-log scale: $G_P = \frac{\left(\frac{dP(t)}{P(t)}\right)}{dt} = \frac{d\ln P(t)}{dt}$ (Eq. (5)). Emerging characteristic time-related events are indicated.

where $a = 0.035$ and $b = 1.52 \times 10^{-4}$.

Consequently, the linear pattern of $G_P(P)$ in Fig 2 may be the result of the linear approximation shown in Eq. 6, the experimental error, and the 'scale compression'.

Fig 4 and Eq. (6) mean that instead of the terminal maximal global population indicated in Fig 2, a permanent rise, with a slowing growth rate, should be expected in the future.

Very recently, the time-related changes in the global population were analyzed for the same set of data as in the given report via the following Super-Malthus equation [84]:

$$P(t) = P_0 \exp\left(r(t) \times t\right) = P_0 \exp\left(\frac{t}{\tau(t)}\right) \tag{8}$$

where the relaxation the time-dependent relaxation time and the time-dependent growth rate were introduced: $\tau(t) = \frac{1}{r(t)}$. For the simple case $r(t) = const$, one obtains the basic Malthus equation. The relaxation time in Eq. (7) allows for estimating the time expected for a hypothetical 50% population rise: $t_{50\%} = \tau \times ln2$.

For the Industrial Revolutions times, starting near the year $t_0 \approx 1700$, regarding the global population $P_0 \approx 0.6 billion$ the linear pattern for the relaxation time changes was noted $\tau(t) \approx a - b(t - t_0)$. The substitution to Eq. (7) led to [84]:

$$P(t) = P_0 \exp\left(\frac{b' \times t}{T_c - t}\right) \Rightarrow P(t) = P_0 \left(1 + \frac{b' \times t}{T_C - t} + \dots\right) \propto \frac{B}{D - t} \tag{9}$$

The analysis of $\tau(t)$ changes in ref. [84], yielded the year $T_C \approx 2226$, which was named the 'critical Dooms-year. Notable that such dynamics appear for the relaxation on approaching critical points in frustrated complex systems (in the meaning of Critical Phenomena Physics [84 and refs. therein]). Omitting higher order terms in the Taylor expansion of the exponential part in Eq. (8), one obtains coincidence with the famous von Foerster Doomsday equation [16,84], recalled in the right-hand part of Eq. (8). Von Foerster et al. [16] formulated the 'hyperbolic' behavior hypothesis via simple empirical analysis of 26 'empirical' global population data from $\sim 400 BC$ till $1958$, which resulted in the 'hyperbolic' anomalous behavior with the 'Doomsday' at $D \approx 2016$ [16]. Such singular, catastrophic behavior attracted broad attention [84 and refs. therein]. Considering Eq. (8) in frames of complex systems dynamics, one can expect finite-value tunneling through $T_C$ time surrounding, then avoiding the infinite singularity [84]. Notable that the 'hyperbolic' von Foerster et al. [16] scaling relation can be coupled to the following reference differential equation:

$$\frac{dP(t)}{dt} = \delta \times [P(t)]^2 \Rightarrow G_P = \frac{1}{P(t)}\frac{dP(t)}{dt} = \frac{dlnP(t)}{dt} = \delta \times P(t) \tag{10}$$

The pattern indicated by Eq. (9) coincides with the linear behavior noted in Fig 2 and in ref. [62] for $G_P$ changes.

Notable, that the time-related singular exponential behavior described by Eq. (8) resembles the pattern developed for complex frustrated and constrained critical dynamics in the *Critical Phenomena Physics* [84]. For systems, the avoided criticality is a common feature. In ref. [84] the empowered exponential Super-Malthus behavior for the global population growth was discussed [84]:

$$P(t) = P_0 \exp(rt)^\beta = P_0 \exp\left(\frac{t}{\tau}\right)^\beta \tag{11}$$

where population growth rate $r = \frac{1}{\tau}$, and $\tau$ is the relaxation time.

 

The results of such portrayal, with related parameters, are shown in Fig 4. Such relation recalls the Weibull distribution for long-time dynamics or Kohlraush-Williams-Watts (KWW) dynamics in complex system physics [84]. The latter links the exponent $\beta < 1$ to the stretched exponential behavior, with the broad distribution of relaxation processes and energy dissipation. For $\beta = 1$ one obtains the basic Malthus dependence, which can be linked to the single, dominant relaxation process and energy conservation for dynamics in the system following the KWW model analysis [84]. It is notable that the results presented in Fig 4 allow for extrapolations, forecasting the global population in the nearest decades. Namely, considering the population slowing down growth trend emerging after the year ~1966, particularly noted in Figs (2) and (3), one obtains $P(2030) \sim 8.9 billion$, $(2050) \sim 11.3 billion$ and $(2100) \sim 20 billion$. Notable that the extrapolation based on the 'compressed' trend obeying before the year ~1968 yields $P \sim 11 billion$ already for the year 2023, whereas the real value, associated with the new, 'stretched' trend, is much lesser: $P \sim 8\ billion$.

## Conclusions

In the recent report by Lehman et al. [62], the multi-parameter Verhulst-type model relation [15] extended by Pearl & Reed [85,86], Volterra [87], and Cohen [88] was implemented for describing global population changes. The success was possible by considering a sequence of $(r, s)$ parameters linked to overcoming subsequent eco-barriers since the Anthropocene onset. Changes in values and signs of these parameters were supported by the discovery of two linear domains for the discrete per capita relative global population change factor $G_P^i(P)$ (Eq. 5), namely: (i) from $10,000 BC$ till $\sim 1962$ with the positive slope and (ii) from $\sim 1962$ till $2010$ with the negative slope.

In the given report, the analytic counterpart of the per capita relative population growth parameter $G_P = \frac{d\ln P(t)}{dt} = \frac{\left(\frac{dP(t)}{P}\right)}{dt}$ is considered. It is implemented for the new set of global population data obtained via numerical filtering of data from different sources. The first view of $G_P(P)$ pattern confirmed the mentioned behavior in ref. [62]. However, the focused view revealed that the first linear domain should be limited to the period $\sim 1000 CE < t < 1966 \pm 3$. It is further argued that the characteristic pattern for $G_P(P)$ changes yield hypothetically optimal pair of $(r, s)$ parameters for describing $P(t)$ changes via the Verhulst Eq. (4), in each domain time-domain indicated above, respectively. However, it does not yield $P(t)$ changes description. On the other hand, the comparison of Eqs. (3) and (5) suggest that the linear behavior of per capita growth rate $G_P(P) = r + sP$ can be considered the validation test for the Verhults equation. Hence, a question arises if the Verhulst-type modeling should be used to describe global population evolution and if the discussed behavior of $G_P(P)$ is not a hallmark of a different model scaling.

Recently, the analysis of $G_P(t)$ was used for such a test focused on portraying global population evolution via two super-Malthus relations, namely Eqs. (7) and (10) are briefly discussed above [84]. They offer a fair portrayal of $P(t)$ global population data and can also be related to the linear behavior of $G_P(P)$, as indicated in Eqs. (6, 8).

In the authors' opinion, the question of the 3rd-path of Verthulst model implementation mentioned above remains. It is related to progressive and self-adaptive changes in the population itself, further renormalizing the system's carrying capacity towards new, lesser needs of the population. It can be called a 'spontaneous self-adaptation of complex 'active-matter population'recalling the language of complex systems physics. It seems to coincide with the sustainable civilization trend, which is dominant nowadays. The authors want to stress the significance of the new path implemented in this report, namely: (i) the application of numerical filtering, which enables the effective use of population data from various sources, (ii) the application of distortion-sensitive and derivative-based transformation of $P(t)$ data, enabling the model-free preliminary insight; it is also the case of per capita global population rate coefficient.

Finally, the authors stress the approach proposed in the given report and ref. [84] can be implemented for arbitrary time-evolving data, from biology and medicine to economic issues. The particular efficiency of such a bottom-up approach matched with the distortions-sensitive analysis can appear when local distortions, also aperiodic, distort or even hide the leading trend.

## Supporting Information

**S1 Appendix. Global population data since Anthropocene onset obtained by collecting data available in refs. [128–135], and subsequently their numerical filtering.**
(DOCX)

## Author contributions

**Conceptualization:** Agata Angelika Sojecka.

**Data curation:** Aleksandra Drozd-Rzoska.

**Formal analysis:** Aleksandra Drozd-Rzoska, Agata Angelika Sojecka.

**Funding acquisition:** Aleksandra Drozd-Rzoska.

**Investigation:** Agata Angelika Sojecka.

**Methodology:** Agata Angelika Sojecka.

**Software:** Aleksandra Drozd-Rzoska, Agata Angelika Sojecka.

**Validation:** Agata Angelika Sojecka.

**Writing – original draft:** Agata Angelika Sojecka.

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
