## [Decision Letter · Decision Letter 0]

4 Jul 2024

PONE-D-24-03938Verhulst Equation and the  Universal  Pattern for the Global Population GrowthPLOS ONE

Dear Dr. Drozd-Rzoska,

Thank you for submitting your manuscript to PLOS ONE. After careful consideration, we feel that it has merit but does not fully meet PLOS ONE’s publication criteria as it currently stands. Therefore, we invite you to submit a revised version of the manuscript that addresses the points raised during the review process.

**ACADEMIC EDITOR:**

Clarify the paper's contribution to the topic and highlight what is new in the abstract or introduction. Increase references in the introduction to better engage with existing literature. Include pictorial illustrations to summarize the story and improve the article.The theoretical section needs to be shorter and include only essential equations. In the introduction, clearly define the motivation, research gap, and research question.Examine similarities between "critical system" and "critical phenomena" in other fields.Use the Theory of Minimum Least Squares (TMLS) to provide additional information on parameters a and b.Apply ARMA, ARIMA models, Neural Networks, or Chaos Theory for prediction purposes, and analyze the system's entropy from the time series data. Improve the theoretical background and explain the study's relevance and contribution to existing knowledge.

We look forward to receiving your revised manuscript.

Kind regards,

Kavikumar Jacob, Ph.D

Academic Editor

PLOS ONE

Journal Requirements:

 [National Center for Science (NCN, Poland), grant ref. 2022/45/B/ST5/04].  

[The work by was supported by the National Center for Science (NCN, Poland), grant ref. 2022/45/B/ST5/04. ]

 [National Center for Science (NCN, Poland), grant ref. 2022/45/B/ST5/04]

Additional Editor Comments:

What specific tools should the authors use to better explain their results?

How can the authors improve the integration of the theoretical background and build a stronger case for the need for their study?

What insights can the authors provide to showcase the novelty and relevance of their work and demonstrate theoretical value and significance within their discipline?

he authors should engage more with existing literature, analyze previous studies, and clearly define the motivation, research gap, and research question in the introduction.

Instead of presenting a list of previous research, the literature review should focus on explaining the relationship between different concepts and theories.

The authors should ensure that the theoretical value and significance of the work are clearly showcased, and the conceptual model is presented in a less complex manner to make a more rigorous contribution to theory.

Reviewers' comments:

Reviewer's Responses to Questions

**Comments to the Author**

1. Is the manuscript technically sound, and do the data support the conclusions?

Reviewer #1: Partly

Reviewer #2: Yes

2. Has the statistical analysis been performed appropriately and rigorously? 

Reviewer #1: No

Reviewer #2: Yes

3. Have the authors made all data underlying the findings in their manuscript fully available?

Reviewer #1: Yes

Reviewer #2: Yes

4. Is the manuscript presented in an intelligible fashion and written in standard English?

Reviewer #1: Yes

Reviewer #2: Yes

5. Review Comments to the Author

Reviewer #1: The paper is very interesting however authors have to use more tools to explain their results. In this version the paper authors present data without to explain the behaviour of Global Population. My comments focus on this problem.

Reviewer #2: In the introduction section, new reference support is very less. The current introduction is partially good but does not seem to engage with existing literature adequately. It is highly recommended that previous studies be analyzed and mentioned.Some pictorial illustrations to summarize the story could be beneficial for improving the article. Authors have made good efforts in developing a good research gap, however, support with the a mix of old and existing studies. Please offer a clear motivation, gap, and research question in the introduction if possible. The introduction could do more to ground the paper's RQ in the debate and the related literature. In the actual version of the manuscript, scant attention is given to a theoretical derivation of the study's RQ and its actual positioning. Authors need to emphasize the novelty and relevance of their investigation by highlighting how the study contributes to the existing body of knowledge after the research question.

Although, the literature review reads more like a list of previous research on various topics rather than a theory section explaining how your different concepts are related. Motivation needs to be improved, and the aim and objectives of the study, its novelty and/or contribution need to be clearly defined. Try to integrate this section better and build a stronger case of the need for your study. I was expecting the authors to start their theoretical background section with the theoretical underpinnings of the study .

-It must include a whole picture of the problem statement and review of critically literature.. Although this paper dealt with interesting phenomena, it did not provide adequate theoretical background and support for the development of its hypotheses. By showcasing the novelty and relevance of your work, you demonstrate theoretical value and significance within your discipline. Some of the main aspects are not properly explained, and others are too broad and not essential from the approach authors have undertaken. The underlying conceptual model seems overly complex; again, benchmarking off papers in leading journals may give some insights into how this might be rendered in a more parsimonious fashion. In turn, this may result in a more rigorous contribution to theory; the author presently skates over theory in a very superficial manner.

6. PLOS authors have the option to publish the peer review history of their article (what does this mean? ). If published, this will include your full peer review and any attached files.

**Do you want your identity to be public for this peer review?** For information about this choice, including consent withdrawal, please see our Privacy Policy .

Reviewer #1: No

Reviewer #2: No

---

## [Author Response · Author response to Decision Letter 1]

17 Aug 2024

Dear Editor,

Attached please find the revised manuscript ‘ Verhulst Equation and the Universal Pattern

for Global Population Growth’, by Agata Angelika Sojecka and Aleksandra Drozd-Rzoska in which all reasonable comments of reviewers have been positively addressed.

Note general supplementations:

1. The paper has been deeply cleaned regarding language.

2. Following reviewers comments the new Figure 4 + related comments at the end of Results & Discussion section

3. Note ca. 40 new references, added to meet reviewers' requirements

4. Note the strongly corrected Abstract and Conclusions, to meet reviewers expectations.

5. Note the report [97] ‘Sojecka, AA, Drozd-Rzoska A. Global population: from Super-Malthus behavior to Doomsday criticality. Scientific Reports. 2024; 14; 9853. https://doi.org/10.1038/s41598-024-60589-3 - which is strictly related to the given paper. It is a complementary paper submitted to Nature-Springer a month later (5th March 2024) after the submission to PlosOne and very quickly accepted after 3 professional and positive opinions, so it appeared in May 2024.

The results from this report are significant for the given report. This report is recalled a few times, since it significantly support the results presented in the given manuscript.

6. Note that the re-submitted report is ‘placed’ in PlosOne template (available in the Net), and follows the template's requirements. All references are corrected according to PlosOne rules.

The Academic Editor (AE) suggested:

1. AE: Clarify the paper's contribution to the topic and highlight what is new in the abstract or introduction. Increase references in the introduction to better engage with existing literature. Include pictorial illustrations to summarize the story and improve the article.

Response: The Introduction has been cleaned and clarified, and the re-written, precise motivation is no given. It is strongly stressed that the report is related to the Verhulst equation portrayal and test of the coupled hypothetical universality of the per-capita relative population growth coefficient. The novelty related to the way of global population data and the distortions – sensitive analysis. In the Introduction there are numerous new references, particularly from 2023-2024, to show the current significance of the topic. Note the new Figure 4 and correction in Figures 2 and 3.

2. AE: The theoretical section needs to be shorter and include only essential equations. In the introduction, clearly define the motivation, research gap, and research question.

Response: the reduction of the number of equations does not agree with sugegstions of reviewers and could make the paper not-clear. Shortening the Introduction is in disagreement with the suggestion of adding significantly more references and report clarification.

The final part of results and discussion has been re-written, and now cognitive challenge/gap and motivation/research questions… are explicitly and precisely presented. See lines 94 – 117.

3. AE: Examine similarities between "critical system" and "critical phenomena" in other fields.

Responce: the explicit link of these names to Critical Phenomena Physics, a commonly known branch of physics and complex systems science, has been recalled a few times. Note particularly the end of the Results and Discussion section.

4. AE: Use the Theory of Minimum Least Squares (TMLS) to provide additional information on parameters a and b.

Response: First note that the Savitzky – Golay filtering routine is better explained in the Methods section, with supporting references. I guess this comment is related to parameters given in Table I, which was determined to use the linear regression routine. It is so standard and popular procedure that its explanation has minimal sense, in my opinion. It is now explicitly stated in the caption of Table I.

5. AE: Apply ARMA, ARIMA models, Neural Networks, or Chaos Theory for prediction purposes, and analyze the system's entropy from the time series data. Improve the theoretical background and explain the study's relevance and contribution to existing knowledge

Response: Note that the theoretical background is well explained now. It follows a new path of global population analysis. It has been introduced, for the first time, in the report mentioned above (in the reference list, it is position 97). In fact, this report in PlosOne could be the first communicate on this cognitive path, but we waited 6 months (!) for opinions (!)

As for ARMA, ARIMA etc… models – this issue is explained in commented at the end Conclusions section. Lines 536-563.

As the profesionalist in complex systems/matter science, I should additionally comment this issue. This suggestion is well beyond the target and motivation of the given report. Second, so fat there are no reasonably valueable report applying such approaches for global population. This comment is related to action with real global population data. There are pure model-theoretical analyses, but they are meaningless without implementation for real data. This lack is not accidental.

Third, the entropy analysis for the time-related non-monotonic global population data is strange. I am not surprised that it has not been done before.

This summary of AE comments and Authors explanations address all issues indicated by Reviewers.

In fact, it was difficult to address all these comment, because their targets were contradictory. It was suggested that the report should be shorter, with simple and straightforward targets and motivations

On the other hand, some specific suggestions were well beyond the current state-of-the-art, and could strongly defocus the report.

Nevertheless, I made the best to meet all reasonable suggestions.

I submitted the basic report a lot of months ago. So now, I will be grateful for the fast final decision: Yes or No.

Thanking for Your troubles

Yours Sincerely

Prof. Aleksandra Drozd-Rzoska

Institute of High Pressure Physics

Polish Academy of Sciences

---

## [Decision Letter · Decision Letter 1]

23 Oct 2024

PONE-D-24-03938R1Verhulst Equation and the Universal  Pattern for Global Population GrowthPLOS ONE

Dear Dr. Drozd-Rzoska,

Thank you for submitting your manuscript to PLOS ONE. After careful consideration, we feel that it has merit but does not fully meet PLOS ONE’s publication criteria as it currently stands. Therefore, we invite you to submit a revised version of the manuscript that addresses the points raised during the review process.

Based on the comments from both reviewers of the manuscript titled "Verhulst Equation and the Universal Pattern for Global Population Growth":

The reviewers acknowledge that the authors have made efforts to address some points raised in the initial review. However, both emphasize that the manuscript, with some significant improvements, has the potential to make a substantial contribution to the field. The primary concerns centre around the originality and contribution of the work. Specifically, the reviewers note that the application of the Verhulst model is not novel, as it has been utilized frequently in prior research. For the paper to make a more substantial contribution, the authors need to articulate their unique insights and contributions clearly. Additionally, while the authors use the TMLS (Two-Mode Logistic System) for modeling, one reviewer questions its suitability for predictive purposes and suggests a comparison with other models, such as ARMA and ARIMA, to validate the results.

Moreover, the reviewers find the theoretical section of the paper too lengthy, recommending that it be streamlined to include only the essential equations. This will improve the manuscript's clarity and focus. Overall, a major revision is needed to address these concerns, improve the justification of results, and refine the theoretical presentation.

We look forward to receiving your revised manuscript.

Kind regards,

Kavikumar Jacob, Ph.D

Academic Editor

PLOS ONE

Journal Requirements:

Additional Editor Comments :

Based on the comments from both reviewers of the manuscript titled "Verhulst Equation and the Universal Pattern for Global Population Growth":

The reviewers acknowledge that the authors have made efforts to address some points raised in the initial review. However, both emphasize that the manuscript, with some significant improvements, has the potential to make a substantial contribution to the field. The primary concerns centre around the originality and contribution of the work. Specifically, the reviewers note that the application of the Verhulst model is not novel, as it has been utilized frequently in prior research. For the paper to make a more substantial contribution, the authors need to articulate their unique insights and contributions clearly. Additionally, while the authors use the TMLS (Two-Mode Logistic System) for modeling, one reviewer questions its suitability for predictive purposes and suggests a comparison with other models, such as ARMA and ARIMA, to validate the results.

Moreover, the reviewers find the theoretical section of the paper too lengthy, recommending that it be streamlined to include only the essential equations. This will improve the manuscript's clarity and focus. Overall, a major revision is needed to address these concerns, improve the justification of results, and refine the theoretical presentation.

Reviewers' comments:

Reviewer's Responses to Questions

**Comments to the Author**

1. If the authors have adequately addressed your comments raised in a previous round of review and you feel that this manuscript is now acceptable for publication, you may indicate that here to bypass the “Comments to the Author” section, enter your conflict of interest statement in the “Confidential to Editor” section, and submit your "Accept" recommendation.

Reviewer #1: (No Response)

Reviewer #3: (No Response)

2. Is the manuscript technically sound, and do the data support the conclusions?

Reviewer #1: Partly

Reviewer #3: Partly

3. Has the statistical analysis been performed appropriately and rigorously? 

Reviewer #1: Yes

Reviewer #3: Yes

4. Have the authors made all data underlying the findings in their manuscript fully available?

Reviewer #1: Yes

Reviewer #3: Yes

5. Is the manuscript presented in an intelligible fashion and written in standard English?

Reviewer #1: Yes

Reviewer #3: No

6. Review Comments to the Author

Reviewer #1: In the revised version of the paper entitled “Verhulst Equation… Population Growth” the authors have taken into account some points of review. However they have to suggest clearly their contribution with this paper. The application of model, which has used many times, is not a contribution worth for publication. Also, they apply the TMLS which is very simple model but not suitable for prediction as others. They have to justify the validity of their results and to discuss in correlation with other models (For example Arma and Arima). Finally, the theoretical section of the paper should shorter. It needs essential equations only.

The paper needs major revision.

Reviewer #3: I have attached a letter of review as a PDF file, or am trying to. It appears that "Upload Review Attachments" is the way to do this, but that is not completely clear on the website.

7. PLOS authors have the option to publish the peer review history of their article (what does this mean? ). If published, this will include your full peer review and any attached files.

**Do you want your identity to be public for this peer review?** For information about this choice, including consent withdrawal, please see our Privacy Policy .

Reviewer #1: No

Reviewer #3: No

---

## [Author Response · Author response to Decision Letter 2]

30 Oct 2024

Dear Editor

Attached please find the second revision of the report ‘ Verhulst-type Equation and the Universal Pattern for Global Population Growth‘. Please note that we decided to make a subtle change in the title, motivated by reviewers' comments. The basic title was ‘ Verhulst Equation and the Universal Pattern for Global Population Growth ‘

We are grateful for the reviewers' comments, which notably influenced the report. We are particularly grateful to Prof. Lehman, the author of the extended Verhulst-type equation, who is the key focus of the report.

We are happy that all reviewers see the cognitive potential of the report.

Regarding opinions of Reviewers #1 and #2

1. Reviewers comment: ‘ The primary concerns centre around the originality and contribution of the work. Specifically, the reviewers note that the application of the Verhulst model is not novel, as it has been utilized frequently in prior research ‘

THE ANSWER: the ‘deep reference’ of the report is the Verhulst equation, but this report is basically related to its qualitative extension by Lehman et al. [PNAS 2021; 118: e2024150118], where the successful global population in the last 12 millennia and the new concept of the per capita population growth was developed.

To stress more this point we decided to a ‘subtle’ change in the title (see above) and add the following sentence in lines (116 - 122): The primary inspiration for this report was the recent report by Lehman et al. [62], which combines the mentioned basic 1st and 2nd cognitive paths for global population studies and considers the Verhulst-type scaling equation associated with the concept proposed by Pearl and Reed [85,86] and further developed by Volterra [87] and Cohen [88]. We stress this issue because such an approach essentially extends the basic Verhulst (including Two-Mode Logistic (TML) approach), often questioned for its suitability for predictive purposes regarding human populations.’

2. Reviewers comment: ‘For the paper to make a more substantial contribution, the authors need to articulate their unique insights and contributions clearly. Additionally, while the authors use the TMLS (Two-Mode Logistic System) for modeling ‘

THE ANSWER: This point is partially explained by the above comment (line 116-122). The key novelty is the introduction of the new ‘bottom–up’ approach, namely (i) numerical filtering of empirical data, (ii) their distortions sensitive analysis exploring the new analytic extension of per-capita relative population growth rate Gp, (iii) the ultimate validation test of the given scaling relations as well identification of domain in which it can be applied. This is the new path for any population data analysis, including global one.

This issue is finally stressed also in conclusions. Prof. Lehman noted this novelty and the possibility of using the proposed methodology beyond global population analysis, for instance, in biology.

3. Reviewers comment: ‘one reviewer questions its suitability for predictive purposes and suggests a comparison with other models, such as ARMA and ARIMA, to validate the results‘

THE ANSWER: This issue is – in detail – explained in lines 95 – 116, with the support of a new reference. Particularly it is indicated that although ARMA/ARIMA approaches are broadly used in business, medicine, or for describing (time-limited!) development or urban centers. Reports regarding global population growth are minimal – because they cannot lead to conclusive results, particularly when considering extreme periods, as in the given report or the paper by Lehman et. al. [PNAS, 2021].

Moreover, introducing such an analysis – despite its fundamental problems – would break the consistency of the work, introducing a qualitatively different problem. It would also significantly increase the size of the report, contrary to the editor's suggestion.

4. General suggestion: ‘Moreover, the reviewers find the theoretical section of the paper too lengthy, recommending that it be streamlined to include only the essential equations’.

THE ANSWER: Note that now the report contains only 10 equations, with re-arrangement not-loosing interpretations. Also some comments have been removed.

However, please note that this requirement does not agree with points (1 – 3 ) above, particularly related to ARMA/ARIMA concept which had to increase the size of the report and the number of references.

Prof. Lehman comments:

Prof. Lehman was very satisfied with the given report and developed a new path concept regarding the extended Verhulst-type modeling in his recent report (PNAS, 2021). He suggested some corrections regarding equations, making them more informative and clear. It has been done. He also suggested some stylistic comments, for instance, minimizing the usage of the word ‘evolution’, which has a different meaning for readers from biology – related communities. Following comments of Prof. Lehman, the following sentence terminates the report: (lines 528-532) ‘ Finally, the authors stress that the approach proposed in the report and ref. [84] can be implemented for arbitrary time-evolving data, from biology and medicine to economic issues. The particular efficiency of such a bottom-up approach matched with the distortions-sensitive analysis can appear when local distortions, also aperiodic, distort or even hide the leading trend.’

In conclusion, we explicitly and positively considered all reviewer comments. We are happy that the cognitive value of the report emerges. Particularly, we are happy and grateful for the opinion of Prof. Lehman, the world-key researcher in population studies nowadays.

With best regards

Agata Angelika Sojecka (Univ. Econom., Katowice, Poland).

Aleksandra Drozd-Rzoska (IHPP PAS, Warsaw, Poland)

---

## [Decision Letter · Decision Letter 2]

7 Jan 2025

PONE-D-24-03938R2Verhulst-type Equation and the Universal  Pattern for Global Population GrowthPLOS ONE

Dear Dr. Drozd-Rzoska,

Thank you for submitting your manuscript to PLOS ONE. After careful consideration, we feel that it has merit but does not fully meet PLOS ONE’s publication criteria as it currently stands. Therefore, we invite you to submit a revised version of the manuscript that addresses the points raised during the review process.

Among the two reviewers' comments, one accepted the manuscript for further processing, and Dr. Lehman addressed various issues and comments about the manuscript listed in the attachment file. 

We look forward to receiving your revised manuscript.

Kind regards,

Kavikumar Jacob, Ph.D

Academic Editor

PLOS ONE

Journal Requirements:

Additional Editor Comments:

One of the reviewer listed some minor corrections.

Reviewers' comments:

Reviewer's Responses to Questions

**Comments to the Author**

1. If the authors have adequately addressed your comments raised in a previous round of review and you feel that this manuscript is now acceptable for publication, you may indicate that here to bypass the “Comments to the Author” section, enter your conflict of interest statement in the “Confidential to Editor” section, and submit your "Accept" recommendation.

Reviewer #1: All comments have been addressed

Reviewer #3: (No Response)

2. Is the manuscript technically sound, and do the data support the conclusions?

Reviewer #1: Yes

Reviewer #3: Yes

3. Has the statistical analysis been performed appropriately and rigorously? 

Reviewer #1: Yes

Reviewer #3: Yes

4. Have the authors made all data underlying the findings in their manuscript fully available?

Reviewer #1: Yes

Reviewer #3: Yes

5. Is the manuscript presented in an intelligible fashion and written in standard English?

Reviewer #1: Yes

Reviewer #3: Yes

6. Review Comments to the Author

Reviewer #1: The authors have taken into account previous reviews and now this version of the manuscript is acceptable for publication.

Reviewer #3: I have uploaded a PDF letter that you should recieve as part of this review. I think you've made some nice progress an don't have so far to go. And I think you know you are working in a very important area!

-- Clarence Lehman

7. PLOS authors have the option to publish the peer review history of their article (what does this mean? ). If published, this will include your full peer review and any attached files.

**Do you want your identity to be public for this peer review?** For information about this choice, including consent withdrawal, please see our Privacy Policy .

Reviewer #1: No

Reviewer #3: No

---

## [Author Response · Author response to Decision Letter 3]

31 Jan 2025

Dear Editor,

Thank you very much for sending the next tour opinion regarding the paper ‘ Verhulst-type Equation and the Universal Pattern for Global Population Growth ‘ (PONE-D-24-03938R2).

One of Reviewers accept the current form of the manuscript. The next Reviewer, Prof. Clarence L. Lehman send a letter with some supplementary comments which should ne also addressed. Finally he expressed a general positive opinion on the report an related research.

I am very for these comments which really make the report better and more in-depth presenting arguments. I am grateful for these comments and opinion, because for me Prof. Lehman is an exceptional world authority in the field of population research and this opinion and the comments contained in it are of great importance to us, also for further research.

Comments notably improved the report and we are very grateful.

Please note: following advices Figures have been improved, by adding description, so they are now ‘more communicative’. for instance for students. The responses to required additional referrnces, also indicated by the reviewer.

Below please find responses to comments, point-by-point.

Reviewer: ‘Line 18. Uses BC here and CE elsewhere. Shouldn't this be unified, such as BCE? I also noticed it later on line 130 and elsewhere.’

Response: It has been dome. In the report only CE and BCE is now used.

Reviewer: ‘Line 22. The difference in dates is not an important issue, but I'm wondering if yours comes up with 1968 versus our 1962 for the transition year because the smoothing that you use takes a little while to turn around? Something to think about perhaps.’

Response: in line22 (Abstract), the ‘approximate’ sign has been introduced. In lines 132-133 the year 1962 instead 1968 has been corrected – since it recalls ref. [64], recalled in the above comment. This issue is now explained via the introduction of the error+comments, Lines 394-396: ‘‘Regarding the crossover year (1968), notable is the error related to three standard deviations and the intersection of two line in Fig.3.’

Reviewer: ‘Line 23. I'm not sure what the phrase (universal?) reference pattern" means. The linearity of these curves is just an approximation, something of a surprising approximation it was to me. The increased population growth following World War II shows up as one of the non-linearities, and of course the major diseases and wars show up as reduced population growth nonlinearities. The generalized population growth model for a single species, or for a group of strongly mutualistic species, would be (1=N)(dN/dt) = r+sN + s2N2 + : : : + siNi + : : : ; as documented by Hutchinson, but just the linear terms approximate us and our domestic mutualists rather well. It also works in various animal populations, going back as far as Allee's Tribolium our beetles.’

Response: the words ‘unique (universal)’ in line 23 has been removed.

Nevertheless, (for future research) the meaning of a simple patter for the global population revealed in ref.[64] and discussed in the given report remains. For me the question arises does it can be a reference for any ‘correct’ scaling equation for the global population growth, which explains the word ‘universal’. But this is the task for the further research and I am grateful for the above comment since it can be a significant indication.

Reviewer: Line 26: ‘It says, \a single pair of growth rate and system resources (carrying capacity) coefficient patterns(r; s): : :" This makes it sound like s is the carrying capacity. The carrying capacity is actually r/s, if s is negative, slowing population growth as the population gets larger. And there is no carrying capacity in the equations if s is positive, which increases the rate of population growth as the population gets larger. (By the way, the reason we named it orthologistic growth is because the asymptote that the population heads toward is vertical when s is positive, whereas it is horizontal at the level of the carrying capacity when s is negative, as in the formulation by Verhulst. In other words, the two asymptotes are orthogonal, hence the term orthologistic.’

Response: to avoid confusion the name (carrying capacity) has been removed, and only ‘resources’ remain’, which correlates with Verhulst basic focused on foods.

The comment is perfect, and once more important for further research, which should follow a strict reasoning pattern.

Reviewer: ‘Line 28: I see you have the term \evolution" there still, which I do believe will cause confusion among biologists, because it suggests the biological evolution of a population, which continually proceeds through time.’

Response: The problem is the meaning and usage of the word ‘evolution’ in different ‘societies’ of researchers. The authors of this report are associated with complex systems physics and socio-economics ‘societies’. To avoid confusion for biologist we changes in lines 28 and 30 the word ‘evolution; to ‘changes’, and a similar correction we made further in other places of the report, if it was possible/

However, please note the title of ref. [99], which shows that using the name ‘evolution’ for complex systems researchers focused on the Global Population is not atypical.

Reviewer: ‘Line 34: Can you really assure that the linearized transformation \yields optimal values of the relevant parameters?" It's certainly yields workable values, but when it smooth out the various plagues, wars, famines, and also baby booms, can we say that the parameters are \optimal?" In all of these methods of fitting the data, the parameters remain approximate.’

Response: Yes, I am sure that the linearized transformation as presented in the Abstract works in this way. It is important that it shows a linear domain in the plot enabling the simple linear regression analysis yield optimal values of parameters with well defined errors. This is not possible for nonlinear routines used for scaling equations, which often has a ‘flat’ minima around obtained values of minima. The situation is even worst if multi-parameter impacts are considered because the problem of their ‘weightings’ appear.

In the statement in line 34, only the reference to explicit ‘mathematic’ background for P(t) data is considered, which should be the basic reference for any other consideration. This issues in/was discussed in the report. See also ref. [114] in Progress in Materials Sciences, the high impact factor journal were publication requires 6 positive reviewers opinions. It shows the unique validation possibilities of the linearized distortions sensitive approach (not for population data but for material engineering, in the given case)/. For more bio & population approach see the recent report [A.A. Sojecka, A. Drozd-Rzoska, S.J. Rzoska, Food preservation in the Industrial Revolution Epoch: Innovative High Pressure Processing (HPP, HPT) for the 21st-Century Sustainable Society, Foods 13, 2024, 3028].

Reviewer: ‘Line 59. I go over some of the things you are mentioning in my ecology classes, but the examples can actually start much earlier |for example with the first stone tools, that could carve hard roots that our ancestors' teeth couldn't readily chew, but could also slice someone's jugular vein, and with _re, which opened a whole new realm for food but also could burn down entire villages. I think it is interesting to talk about such challenges, as you are doing. ‘

Response: In this work we wanted to focus mainly on the Industrial Revolution, which transformed the world in an extraordinary way, also leading to qualitative changes in the growth pattern of the global population. The examples given in this commentary are inspiring, and may be worth developing in further work from pre-historical times through the extraordinary changes in the Neolithic and at the border of the Bronze Age, in correlation with environmental and ecological constraints. This is an issue worth further research, especially in the context of recent and surprising discoveries of Neolithic urban complexes, e.g. in present-day Turkey Republik.

Reviewer: ‘Line 96: The term \optimal solution" arises again. And to make it less controversial, and still accurate, you might consider saying something like \can form an excellent solution." Or if you want readers not to think you are marketing an idea, you could say \worthy solution," or even more modestly, as I mentioned earlier, workable solution." If you seem to be pushing an idea, rather than just presenting it for evaluation, that can make some readers more skeptical, I think.’

Response: Please see the name in Lines 95/96: ‘workable solution’.

This comment is a perfect advise.

Reviewer: ‘Line 129. I notice you have the population growth the way we write it in our papers, and also in our textbook, in the form of (1/P)(dP/dt), but earlier as (dP(t)/P(t)/dt, on line 21. Of course they're equivalent, but when I saw your alternative form, I was wondering if that might be easier for students to understand. I will have to try it both ways and see. I don't know if there's anything here that should be changed, but I just thought I would bring this up.’

Response: Please note the supplemented Line 129, which shows the explicit discrete counterpart of the analytic form of G_P parameter. I think it can help ;less experienced’ readers (Lines 128/129): ’… G_P^i=(1⁄P_i )(〖�P〗_i⁄(�t_i ))=[((〖�P〗_i⁄P_i ))⁄(�t_i )] vs. P_i, where …’

Reviewer: ‘Related to a comment from my previous review, not everyone agrees on what should be called the Anthropocene, so perhaps you should say something like \extending from the time that we are considering the onset of the Anthropocene.’

Response: Please see the correction in lines 137-138: ‘This report focuses on the meaning of this exceptional pattern of G_P for global population changes in the period extending from the time that we are considering the onset of the Anthropocene.’

Reviewer: ‘Line 163. Just a little point, why not write it as P(t) = P0ert? Isn't that a more standard mathematical form, and also avoids the problem of whether the `exp' should be italicized or not?’

Response: Please see the current form of Line 163/164:

dP(t)/dt=rP(t) � P(t)=P_0 e^rt=P_0 exp(rt) (1)

which explains the issue for readers.

It shows the link between both forms of presentations. Nevertheless, in the subsequent part of the report the notation ‘ ‘ is used since it follow the dominant way of presentation in complex systems science and complex systems physics, and in our opinion offers a more clear insight. Following such notation a reference: italics for’ exp(rt) … is obligatory.

Reviewer: ‘Line 175. I think you mean K = -r/s, with a minus sign before the quotient. Otherwise, with a positive r, the right-hand term in Equation 2 will run away and poke through innity along a vertical asymptote, orthologistically, rather than leveling off along a horizontal asymptote as needed for logistic growth.’

Response: it has been corrected (see line 176)

Reviewer: ‘There is a well-developed ecological theory formulated over the past _fifty years or so by David Tilman and others that could be examined and mentioned here. It concerns limited resources interacting with population growth and leads to carrying capacities determined by resource needs. This theory was first previewed by Volterra.’

Response: Please note the new sentence in Lines 191-194:

‘Worth recalling is the model developed by Tilman [94,95] and followers [96], which discussed resources interacting with population growth, which indicates carrying capacities determined by resource needs.

It was associated with 3 new references [94,95,96]. ‘

Reviewer: ‘Line 192I noticed you mention Easter Island as an example. I have recently read that populations there did not proceed as thought, but I haven't looked into it at all. I noticed you don't cite a reference there. It could be something to look into more carefully.’

Response: Please note supplementations answering this comment, in Lines 195-201:

‘Recently, the Verhulst-type pattern has been shown for human population changes on Easter Island (Rapa Nui), the Pacific island, located well remote from other islands and the South America mainland [94]. It is worth mentioning that recent studies have shown that this previously dominant picture, linking ecological constraints and population changes, was substantially changed by research showing the devastating impact of contact with European sailors and later marauders who enslaved people on the South American mainland [97,99].

It was associated with 2 new references [97,99].

15. Reviewer: ‘Line 194 and forward. I have not tried to think about the historical topics you are mentioning here. It's a little beyond my range of knowledge. Some other reviewer would be better for that.'

Response: The above comment, related to (P.14) response this question/problems – particularly when taking into account added references from Nature and Science Advances (both 2024), written in a nice, even popular, way.

15. Reviewer: Line 249. This idea that human population growth follows a logistic equation with carrying capacities con-tinually increasing, to allow the growth curve of 1=N dN/dt versus N to be a line of positive slope, seems tome to be an attempt by modellers of human population during the 20th century to work with a single-species ……..If you'd like to communicate more about this by email, or perhaps by Zoom, I would be happy to explore this further. I think it is a fundamentally

important topic to work through.’

Response: It is a a very interesting comment, not related to the supplementation in the report but to opinions exchanges and discussions. We are very happy to this, and we will contact as soon as the given paper appears. Earlier could be a bit unfair for reviewing rules.

15. Reviewer: ‘Line 258 through 265. in reviewing our paper and your explanation about it, may I suggest the following somewhat modi_ed wording: \Very recently, Lehman et al. [62] have developed these concepts by considering global population growth in terms of three successive bio-ecological levels: (1) interactions with predators, (2) interactions with prey, and (3) intraspeci_c interactions. Global population changes at each level are governed by level-dependent ecological coefficients (ri; si), i=1; 2; 3. These led to population discontinuities progressively separating (i) a primordial phase, where pre-human ancestors interacted with their environment as other animals do, (ii) a mastery of tools, fire, and specialization phase, (iii) an agricultural phase, andfinally (iv) a present controlled-fertility phase. Parameters for population growth changed at each discontinuity. These were implemented for : : :" I would say \ecological coefficients" rather than \Verhulst coefficients" …

Response ‘ Exactly the above description of the (fantastic!) ref. [62] has been including in the report. They are Lines 268-276 in the current version of the reports.

This resume is really better. Thank You.

Reviewer: ‘Line 264. Included in the above, you have \_les" in \tools, _les, and specialization" It should be fire."’

Response: It was a misprint. It has been corrected.

Reviewer: ‘Line 304 through 394. Again, I haven't checked the historical discussion or conclusions here. I did check the Figures though. Also, I see an AD in there..,’

Response: We are sure for the (very soft) historical discussion. Although we do not specialize in history, this branch of knowledge is more than a hobby for us.

We went through the paper and now only CE and BCE appears.

Reviewer: ‘Line 400. I haven't checked the parameters in the table in detail,…. So I would say this table definitely needs a little more work’

Response: please see the corrected Table and the discussion supplement in lines: . Note that values of coefficients are directly taken from the linear regression fit, including the error. We stress in the text that their substitution to the Verhulst equation do nit yield P(t) portrayal in the 1st or the 2nd period. In cam mean that the derivative equation leading to the given plot GP(P) vs. P offers a nice general description for

---

## [Decision Letter · Decision Letter 3]

26 Feb 2025

PONE-D-24-03938R3Verhulst-type Equation and the Universal  Pattern for Global Population GrowthPLOS ONE

Dear Dr. Drozd-Rzoska,

Thank you for submitting your manuscript to PLOS ONE. After careful consideration, we feel that it has merit but does not fully meet PLOS ONE’s publication criteria as it currently stands. Therefore, we invite you to submit a revised version of the manuscript that addresses the points raised during the review process (see Prof Lehman 's comments on the third revised version of your paper)..

We look forward to receiving your revised manuscript.

Kind regards,

Serge Svizzero, Ph.D

Academic Editor

PLOS ONE

Journal Requirements:

Additional Editor Comments :

Please consider Prof Lehman's comments on the third revision of your paper.

Reviewers' comments:

Reviewer's Responses to Questions

**Comments to the Author**

1. If the authors have adequately addressed your comments raised in a previous round of review and you feel that this manuscript is now acceptable for publication, you may indicate that here to bypass the “Comments to the Author” section, enter your conflict of interest statement in the “Confidential to Editor” section, and submit your "Accept" recommendation.

Reviewer #1: All comments have been addressed

Reviewer #3: (No Response)

2. Is the manuscript technically sound, and do the data support the conclusions?

Reviewer #1: Yes

Reviewer #3: Yes

3. Has the statistical analysis been performed appropriately and rigorously? 

Reviewer #1: Yes

Reviewer #3: Yes

4. Have the authors made all data underlying the findings in their manuscript fully available?

Reviewer #1: Yes

Reviewer #3: Yes

5. Is the manuscript presented in an intelligible fashion and written in standard English?

Reviewer #1: Yes

Reviewer #3: Yes

6. Review Comments to the Author

Reviewer #1: The paper has been accepted in the present form since there are improvements compared to the previous version.

Reviewer #3: You're getting there! I found a couple of mistakes in the equations that have to be fixed, described in my attached letter.

7. PLOS authors have the option to publish the peer review history of their article (what does this mean? ). If published, this will include your full peer review and any attached files.

**Do you want your identity to be public for this peer review?** For information about this choice, including consent withdrawal, please see our Privacy Policy .

Reviewer #1: No

Reviewer #3: No

---

## [Author Response · Author response to Decision Letter 4]

16 Mar 2025

Dear Editor,

Attached please find the revised manuscript ‘Verhulst-type Equation and the Universal Pattern for Global Population Growth ‘ ref. PONE-D-24-03938R3

It is the 3rd revision of the manuscript, we hope the last revision terminates 14th month lasting cognitive ‘adventure’ associated with the submission to PlosONE.

Two reviewers supported the publication (YES). The 3rd Reviewer – Prof. Lehman, suggested some additional corrections.

We are very grateful to Prof. Lehman for his inspiring comments and suggestions. They led to a significant improvement in the report and progressed understanding and explaining puzzling issues for the given problem.

There are two essential comments of Prof. Lehman

1. The first one is related to the correctness of some equations, particularly related to the chapter ‘ Remarks on Malthus and Verhulst equations‘

2. The second one is related to the general name or more – consequences – of ‘ multi-parameter Verhulst-type model ‘

Response:

• All equation are now re-tested. Note that the notation referred to the basic (standard) Verhulst presentation, and the reference definition Gp =r – sP is used for the whole paper.

• Please note that the chapter Remarks on Malthus and Verhulst equations has been re-written and also presents those facts in a way that can be interesting for students in following the topic step-by-step

• Professor Lehman's breakthrough ‘empirical’ finding is related to Fig. 3 in ref. [62], portrayed by lines with different slopes, which means signs of the resources s parameter. The way to the solution of this grand mystery is now commented/explained at the end of the ‘Remarks…..’ chapter using the Cohen approach.

• the next issue and for us grand mystery – but also undoubtful ‘emprirical’fact/ reference is the fair portrayal in a ‘multi-parameter’ Verhulst patter.

Please, note the slightly supplemented Figure 2 (with comments below the figure) showing the polynomial portrayal and the comment below.

The report has also been ‘cleaned’ in depth. We hope that this supplementation/correction finishes the 14th months' lasting story associated with the submission to PlosONE. Some new questions have appeared over this long time, but they can only be the target of further studies and reports.

Agata Angelika Sojecka and Aleksandra Drozd-Rzoska

---

## [Editor Report · Decision Letter 4]

4 Apr 2025

Verhulst-type Equation and the Universal  Pattern for Global Population Growth

PONE-D-24-03938R4

Dear Dr. Drozd-Rzoska,

We’re pleased to inform you that your manuscript has been judged scientifically suitable for publication and will be formally accepted for publication once it meets all outstanding technical requirements.

Kind regards,

Serge Svizzero, Ph.D

Academic Editor

PLOS ONE
---

## [Editor Report · Acceptance letter]

PONE-D-24-03938R4

PLOS ONE

Dear Dr. Drozd-Rzoska,

I'm pleased to inform you that your manuscript has been deemed suitable for publication in PLOS ONE. Congratulations! Your manuscript is now being handed over to our production team.

Kind regards,

on behalf of

Pr. Serge Svizzero

Academic Editor

PLOS ONE